Inhibition of SUV39H1 reduces tumor angiogenesis via Notch1 in oral squamous cell carcinoma

Chen Yan 1 2
Weng Xiuhong 1
Zhang Chuanjie 3
Wang Simin 1
Wu Xuechen 1
Cheng Bo 1 chengbo@znhospital.cn
1 Department of Stomatology, Zhongnan Hospital of Wuhan University , Wuhan , China
2 Department of Stomatology, Renmin Hospital of Wuhan University , Wuhan , China
3 Department of Children Health Care, Wuhan Children’s Hospital (Wuhan Maternal and Child Healthcare Hospital), Tongji Medical College, Huazhong University of Science and Technology , Wuhan , China
Song Jian
Electronic publication date: 2024 Apr 19
Publication date: 2024
Volume: 12
Electronic Location ID: e17222
Received 2023 Apr 28; Accepted 2024 Mar 20
Copyright: © 2024 Chen et al.
Copyright year: 2024
Copyright holder: Chen et al.
License: This is an open access article distributed under the terms of the Creative Commons Attribution License, which permits unrestricted use, distribution, reproduction and adaptation in any medium and for any purpose provided that it is properly attributed. For attribution, the original author(s), title, publication source (PeerJ) and either DOI or URL of the article must be cited.
License URL: https://creativecommons.org/licenses/by/4.0/

Keywords: SUV39H1, Notch1, Tumor angiogenesis, Oral squamous cell carcinoma

Funding: The authors received no funding for this work.

==============================
Targeting tumor angiogenesis is an important approach in advanced tumor therapy. Here we investigated the effect of the suppressor of variegation 3–9 homolog 1 (SUV39H1) on tumor angiogenesis in oral squamous cell carcinoma (OSCC). The GEPIA database was used to analyze the expression of SUV39H1 in various cancer tissues. The expression of SUV39H1 in OSCC was detected by immunohistochemistry, and the correlation between SUV39H1 and Notch1 and microvascular density (MVD) was analyzed. The effect of SUV39H1 inhibition on OSCC was investigated in vivo by chaetocin treatment. The migration and tube formation of vascular endothelial cells by conditioned culture-medium of different treatments of oral squamous cell cells were measured. The transcriptional level of SUV39H1 is elevated in various cancer tissues. The transcription level of SUV39H1 in head and neck squamous cell carcinoma was significantly higher than that in control. Immunohistochemistry result showed increased SUV39H1 expression in OSCC, which was significantly correlated with T staging. The expression of SUV39H1 was significantly correlated with Notch1 and CD31. In vivo experiment chaetocin treatment significantly inhibit the growth of tumor, and reduce SUV39H1, Notch1, CD31 expression. The decreased expression of SUV39H1 in OSCC cells lead to the decreased expression of Notch1 and VEGF proteins, as well as the decreased migration and tube formation ability of vascular endothelial cells. Inhibition of Notch1 further enhance this effect. Our results suggest inhibition of SUV39H1 may affect angiogenesis by regulating Notch1 expression. This study provides a foundation for SUV39H1 as a potential therapeutic target for OSCC.

Introduction

Oral squamous cell carcinoma is one of the leading causes of death in cancer patients worldwide (Sung et al., 2021). Although advanced multimodal treatment strategies have improved prognosis and survival in patients with OSCC, the 5-year survival rate for patients with advanced OSCC remains around 50% (Ren et al., 2020). In the process of cancer progression, tumor angiogenesis highly promotes tumor growth and metastasis, and inhibition of angiogenesis has long been a focus of tumor therapy research (Danlos et al., 2023).

SUV39H1, the classical human histone lysine methyltransferase. SUV39H1 introduces di- and trimethylation at histone H3 lysine 9 (H3K9) and plays important roles in the maintenance of heterochromatin and gene repression. It consists of a catalytically active SET domain and a chromodomain, which binds H3K9me and has roles in enzyme targeting and regulation (Pace et al., 2018; Padeken, Methot & Gasser, 2022). SUV39H1 is involved in the regulation of growth and development, diabetes, immune defense, vascular endothelial dysfunction and other processes (Costantino et al., 2019; Niborski et al., 2022; Santos-Barriopedro et al., 2018; Shen et al., 2021). The role of SUV39H1 in tumors is unclear. SUV39H1 may be a tumor suppressor, involved in the occurrence and development of various types of tumors by promoting cell senescence and inhibiting genes required for cell proliferation (Chu et al., 2020). SUV39H1 may also play a carcinogenic role in some tumor types (Morgan & Shilatifard, 2020). The study of SUV39H1 in OSCC has not been reported. A comprehensive understanding of the role of SUV39H1 in OSCC is conducive to the development of new therapeutic strategies to delay the progression of OSCC.

Angiogenesis is closely related to tumor size (Fukumura et al., 2018). Notch1 is involved in cell fate determination in the development and maintenance of tissue homeostasis and is an important regulator of tumor angiogenesis (Lu et al., 2023). In OSCC, Notch1 up-regulates VEGF expression and promotes tumor angiogenesis (Cierpikowski, Lis-Nawara & Bar, 2023). In osteoarthritis, Notch1 is involved in angiogenesis in the synovium of the joint by regulating HIF1ą (Gao et al., 2013). As far as we know, however whether SUV39H1 is related to tumor angiogenesis has not been reported in the literature.

Here, we investigated the role of SUV39H1 in OSCC. The transcription level of SUV39H1 was increased in various cancer tissues, and the transcription level in head and neck squamous cell carcinoma tissues was significantly higher than that in control tissues. The expression of SUV39H1 increased in OSCC, which was significantly correlated with T stage. The expression of SUV39H1 was significantly correlated with Notch1 and MVD. In vivo experiments, inhibition of SUV39H1 reduce the expression of Notch1 and CD31. The decreased expression of SUV39H1 in OSCC cell lines leads to the decreased expression of Notch1 and VEGF proteins, and the inhibition of Notch1 further enhance this effect.

Materials and Methods

Statement of ethics

The study was conducted in accordance with the Declaration of Helsinki, and approved by the Ethics Committee of Zhongnan Hospital of Wuhan University (protocol code WDRY2021-KS019 and date of approval) for studies involving humans. The animal study protocol was approved by the Ethics Committee of Zhongnan Hospital of Wuhan University (protocol code ZN2022085 and date of approval).

Patients

OSCC and matched paracancer tissues were diagnosed by two independent pathologists in the Department of Pathology of Zhongnan Hospital of Wuhan University in accordance with the 2006 WHO grading system. This study included 80 OSCCs and 40 paired paracancer tissues. Please refer to Table 1 for the general information of the patient.

Table 1 The general information of the patient.

Characteristics	OSCC (n = 80)	
Age, mean (y)	56.6	
Gender No.		
Male	72	
Female	8	
Tobacco smoking, No.		
Yes	19	
No	61	
Alcohol drinking, No.		
Yes	39	
No	41	
Tumor site, No.		
Tongue	35	
Buccal	29	
Others (gingiva/lips/palates)	16	
T stage, No.		
I+II	35	
III+IV	45	
Pathological grade, No.		
Well differentiated	48	
Moderately/Poorly differentiated	32	
Lymph node metastasis		
No	58	
Yes	22	

Reagents, antibodies, cell lines and cultures

N-(N-(3, 5-difluorophenyl-l-propyl))-S-phenylglycine t-butyl ester (DAPT, gamma-secretase inhibitor, inhibits the cleavage of Notch1) was obtained from Sigma-Aldrich (St. Louis, MO, USA). Chaetocin (SUV39H1 inhibitors; catalog No. S8068) from Selleck China (Shanghai, China). Antibody SUV39H1 (catalog number: AV32470) was purchased from Sigma-Aldrich, Notch1 (catalog No. 3608), CD31 (catalog No. 3528), and VEGF (catalog No. 50661) from Cell Signaling Technology (Danvers, MA, USA). Antibody SUV39H1 (catalog number: 28074-1-AP) was purchased from Proteintech (WuHan, Hubei province, China). The Human OSCC cell line (Cal27) was purchased from ATCC. Cal27 cells were cultured in DMEM medium (Hyclone, Logan, UT, USA) containing 10% fetal bovine serum (FBS; Gibco, Carlsbad, CA, USA) in a humid atmosphere of 37 °C, 95% air, and 5% carbon dioxide.

Immunohistochemistry

Simply, slices are dewaxed in xylene, rehydrated in graded ethanol and double distilled water, and then heat-induced antigen extraction. After the primary antibody (SUV39H1 1:200, Notch1 1:100, CD31 1:200, Ki67 1:2,000) was incubated at 4 °C overnight, the secondary antibody was incubated at room temperature for 1 h. Image-Pro Plus 6.0 (Media Cybernetics, Inc., Rockville, MD, USA) was used to calculate the density determination. CD31 labeled microvessels, counting capillaries and microvessels in tissue sections. A single brown endothelial cell or cluster of endothelial cells was counted as one vessel, but vessels with thick muscle layer and lumen area larger than 8 red blood cell diameter were not counted. The counting method was to first scan the whole section with a 10*10 low-power lens to determine the field of view with the densest microvessels, and then count all the stained microvessels within the field of view with a high-power lens. The average of the three field counts was taken as the number of microvessels in the cuticle. The tissue sections were scored according to the staining degree (0–3 corresponding to negative staining, light brown, dark brown) and positive range (1–4 corresponding to 0–25%, 26–50%, 51–75%, 76–100%) under the optical microscope. The two scores are added together and compared.

Animal experiments

The in vivo study was approved and supervised by the Animal Care and Use Committee of Wuhan University, and was conducted in accordance with the NIH Guidelines for the Care and Use of Laboratory Animals. Female BALB/c nude mice without thymus (18–20 g; 6–8 weeks of age) were obtained from Wuhan SLAC Laboratory Animal Co., Ltd. (Wuhan, China). At the Laboratory Animal Center of Wuhan University, animals are kept in pressurized ventilation cages, according to the institution’s regulations (five animals per cage). The mice were placed in a suitable sterile cage with a filter lid in the Laboratory Animal Center of Wuhan University and fed and watered at will. Mice were maintained on a 12-h light/dark cycle. When the cell growth was exponential, CAL27 cells (4 × 106, 0.2 ml serum-free medium) were injected subcutaneously into the flank of mice.

The mice were randomly divided into two groups (five animals per group), give Chaetocin (0.25 mg/kg, intraperitoneally (i.p.), once a day; n = 5) or normal saline (control, 100 ul i.p once a day; n = 5) 3 weeks of infusion. Tumor growth was determined by measuring tumor size three times a week. The tumor volume was determined by the formula (width2 × length)/2. The mice were euthanized by cervical dislocation method at a specified time point or maximum diameter of tumor more than 2 cm. The tumors were collected for immunohistochemical analysis. The raw data are provided in the Supplemental Material.

RNA interference experiment

Briefly, CAL 27 cells were seeded in 6 cm petri dishes and grew to 80%. SUV39H1 siRNA was were purchased from Shanghai Genechem Co., LTD (Shanghai, China) and transfected with Hiperfect transfection reagent (Qiagen, Hilden, Germany) according to the manufacturer’s instructions. Western blot confirmed that at a specific time (36 h), the knockout efficiency of SUV39H1 protein was reduced by at least 90%. SUV39H1 siRNA forward 5′-CGUGGAUUGUCUCAGGGAATT-3′; Downstream 5′-UUCCCU GAGACAAUCCACGTG-3′.

Conditioned medium, transwell assay and tube formation

The treated CAL27 cells (SUV39H1 siRNA or/and DAPT) were washed twice with phosphate buffer (PBS). Endothelial basal medium (EBM; Lonza, Walkersville, MD, USA) with serum removed was continued to grow for 24 h. The cleared supernatant was collected as conditioned medium (CM) and stored at −80 °C.

HUVECs cells were suspended in six-well culture plates, and grown to 90% confluence. The Transwell Boyden chamber (Corning Life Sciences, Corning, NY, USA) was used to measure migration of endothelial cells. Starved HUVECs cells (5 × 104 cells/well) incubated in 100 μl of ECM were placed in the upper wells, whereas CM, as a chemo-attractant, was added to the lower wells. Then, cells were incubated for 12 h at 37 °C. HUVECs on the upper surface were carefully scrubbed off using a cotton swab, and cells adhering to the lower membrane were fixed with 4% formaldehyde, stained by crystal violet, and observed under microscope. The rat tail glue was configured first: rat tail glue was composed in the ratio of A:B:C = 9:1:1. Liquid A was 0.02N acetic acid plus 50× Collagen solution (0.02N acetic acid 7.8 μl + Collagen 130 μl + H2O 6.3 ml), and liquid B was 10 × Eagle’s medium solution. C is a 2% sodium carbonate solution. Rat tail gum solution was added to 24-well plate, and the volume of 500 μl/well was added. The solution was set at 37 °C for 1 h, and after solidified, it was washed gently with PBS. HUVEC was inoculated with 5 × 104 cells per well. After several hours of culture, the cells were adherent to the wall. Ordinary medium was replaced with conditioned medium collected after treatment. This should be stopped after 24 h. The supernatant was removed and discarded with gum then gently washed three times with PBS. Then, 4% paraformaldehyde should be used to fix for 10 min; wash with water and then gently wash with PBS. The culture plates were placed under an inverted microscope, and six fields of view were randomly selected to take photos to observe and analyze the number of tubes.

Tumor genome atlas (TCGA) analysis

Data on target gene transcription in different cancer patients were obtained from the GEPIA database (GEPIA http://gepia.cancer-pku.cn/).

Western blot assay

Briefly, CAL27 cells were treated with the indicated concentrations in DMEM containing 2% FBS for 24 or 36 h. Then the cells were lysed, and the total protein was separated using 10% SDS-polyacrylamide gel electrophoresis and transferred onto polyvinylidene fluoride membranes (Millipore Corporation, Billerica, MA, USA). The immunoblots were cultured overnight at 4 °C with the corresponding primary antibodies in blocking solution, followed by incubation with horseradish peroxidase-conjugated secondary antibody (Pierce Chemical, Rockford, IL, USA). Then, the blots were developed with a West Pico ECL kit.

Statistical analysis

Statistical analysis was performed using Graphpad Prism software (GraphPad, Inc., La Jolla, CA, USA), and the data were expressed as mean ± mean standard error (SEM). Two-way ANOVA analysis was used for analyzing differences between animal treatment results. One-way analysis of variance (ANOVA) and student t test were used to analyze the differences with the control group and among each group. The relationship between SUV39H1, Notch1 and CD31 expression was analyzed statistically by Pearson with two tails.

Results

The transcription of SUV39H1 is increased in head and neck squamous cell carcinoma, and the expression of SUV39H1 in OSCC is correlated T stage.

Compared with the corresponding normal tissue (Fig. 1A), SUV39H1 were increased in bladder urothelial carcinoma (BLCA), breast infiltrating carcinoma (BRCA), cholangiocarcinoma (CHOL), colorectal adenocarcinoma (COAD), esophageal carcinoma (ESCA), head and neck squamous cell carcinoma (HNSC), hepatocellular carcinoma (LIHC), lung adenocarcinoma (LUAD), lung squamous cell carcinoma (LUSC), and kidney hyaluria Increased transcription in cell cancer (KIRC), rectum adenocarcinoma (READ), gastric cancer (STAD), and endometrial cancer (UCEC). However, SUV39H1 transcription is reduced in thyroid cancer (THCA). Immunohistochemical staining was used to further study the protein expression of SUV39H1 in OSCC, and it was found that the expression of SUV39H1 in OSCC was significantly higher than that in paracancer tissues (Figs. 1B and 1C). After analysis, it was found that the expression of SUV39H1 in squamous cell carcinoma at T1+T2 was significantly lower than that in squamous cell carcinoma at T3+T4, suggesting that SUV39H1 may play an important regulatory role in the progression of OSCC (Fig. 1D).

Figure 1 The transcription of SUV39H1 in various cancer tissues and the expression of SUV39H1 in oral squamous cell carcinoma were elevated.

(A) Transcription of SUV39H1 in various cancer tissues. (B) Immunohistochemical detection of SUV39H1 expression in oral squamous cell carcinoma and paracancer tissues. (C) Semi-quantitative analysis of SUV39H1 expression in oral squamous cell carcinoma and paracancer tissue. (t test). (D) SUV39H1 expression in different T stages of OSCC. (t test *p < 0.05, ***p < 0.001, bar: mean with SEM).

Correlation between SUV39H1, Notch1, and MVD expression in OSCC

We studied the correlation between SUV39H1, Notch1 and CD31 protein expression by immunohistochemical staining. As showed in Fig. 2A, SUV39H1 was mainly expressed in the nucleus. Notch1 is mainly expressed in the cell membrane, and CD31 is present in cell membranes, cell junctions, and blood vessels. Tissue scores of SUV39H1, Notch1 and MVD were calculated, and then correlation analysis was performed for each of the two proteins selected. Our results showed a significant positive correlation between SUV39H1 and tissue scores of Notch1 and MVD (Figs. 2B and 2C). In addition, Notch1 expression in OSCC was positively correlated with MVD (Fig. 2D). We investigated the correlation between NOTCH1 and CD31 expression in head and neck squamous cell carcinoma in the GEPIA database and found that the expression of NOTCH1 was positively correlated with that of CD31 (Fig. S1). It is worth noting that the correlation between SUV39H1 and NOTCH1 and MVD is weak, which may indicate that SUV39H1 may not be directly involved in tumor angiogenesis in OSCC. This needs to be further clarified in future studies.

Figure 2 Correlation between SUV39H1, Notch1, and MVD expression in OSCC.

(A) A typical image shows SUV39H1, Notch1, and CD31 expression in OSCC. (B) The correlation analysis of SUV39H1 and Notch1 human OSCC (n = 80). (C) The correlation analysis of SUV39H1 and MVD human OSCC (n = 80) and (D) The correlation analysis of MVD and Notch1 human OSCC (n = 80).

Chaetocin attenuates OSCC xenograft tumor growth in nude mice

We used Chaetocin treatment ectopic xenograft tumor derived from CAL27 cells to further determine SUV39H1 function in OSCC. Chaetocin is used to inhibit SUV39H1 function. As shown in Figs. 3A–3D, compared with the control group, chaetocin obviously inhibited tumor growth. Treatment doses of Chaetocin has no obvious toxicity in mice (Fig. 3E). These results suggest that Chaetocin may effectively inhibit tumor growth in OSCC.

Figure 3 Chaetocin attenuates OSCC xenograft tumor growth in vivo.

(A) Image of tumor in Chaetocin treatment group and control group. The red circle shows the location and size of the tumor. (B) Tumor size in the treatment and control groups at the end of the experiment. (C) Tumor growth curves of treatment group and control group. (D) Tumor weight in treatment and control groups. (E) Body weight changes of nude mice in the treatment group and the control group after the in vivo assay (**p < 0.01, ***p < 0.001, bar: mean with SEM).

Chaetocin inhibit tumor angiogenesis in OSCC

After the animal experiment, we collected the transplanted tumor tissue. Immunohistochemical study showed that in the groups treated Chaetocin, SUV39H1, Notch1, CD31, Ki67 staining intensity is lower than the control group (Fig. 4A). We calculated the SUV39H1, Notch1, MVD group score. In OSCC, SUV39H1, Notch1, MVD, Ki67 was significantly lower after Chaetocin treatment (Fig. 4B).

Figure 4 Chaetocin inhibit tumor angiogenesis in OSCC.

(A) Typical immunohistochemical images of SUV39H1, Notch1, CD31 and Ki67 in the treatment and control groups in xenograft tumors. (B) Semi-quantitative analysis of SUV39H1, Notch1, MVD and Ki67 expression in treatment group and control group (**p < 0.01, ***p < 0.001, bar: mean with SEM).

SUV39H1 affects tumor angiogenesis by regulating Notch1

We constructed SUV39H1siRNA and examined its inhibitory effect on SUV39H1 expression (Fig. 5A). As shown in Fig. 5B, SUV39H1siRNA reduced HUVEC migration and tube formion capacity compared with control medium. Similar results were obtained for DAPT treatment. SUV39H1siRNA as well as DAPT treatment further reduced HUVEC migration and tube formion ability (Figs. 5C and 5D). SUV39H1siRNA decreased the expression of Notch1 and VEGF. DAPT decreased the expression of VEGF, but had little effect on the expression of SUV39H1. After SUV39H1siRNA and DAPT were treated simultaneously, the expressions of Notch1 and VEGF were further decreased (Fig. 5E). As shown in Fig. 5F, after treatment with Chaetocin and DAPT, the gray scale of SUV39H1, NOTCH1, and VEGF strips was lower than that after Chaetocin or DAPT alone treatment. The results of Chaetocin treatment were similar to those of SUV39H1 siRNA treatment.

Figure 5 SUV39H1 affects tumor angiogenesis by regulating Notch1.

(A) SUV39H1 siRNA inhibited the expression of SUV39H1 protein. (B) The migration and tube formion ability of HUVECs were different under different conditioned medium. (C) Quantitative of transwell assay. (D) Quantitative of tube formation assay. (E) The expression levels of SUV39H1, Notch1 and VEGF proteins were analyzed by Western blot assay after treatment withSUV39H1 siRNA, DAPT (10 μM for 48 h), and the combination of DAPT and SUV39H1 siRNA. (F) The expression levels of SUV39H1, Notch1 and VEGF proteins were analyzed by Western blot assay after treatment with Chaetocin, DAPT, and the combination of DAPT and Chaetocin (1 μM for 48 h) (**p < 0.01, ***p < 0.001, compared with control group or scramble group, #p < 0.05, ###p < 0.001, compared with SUV39H1 siRNA group or DAPT group, bar: mean with SEM, All cell phenotypic effects were biologically repeated three times).

Discussion

SUV39H1 seems to have different functions in different tumors. In cervical cancer, SUV39H1 knockout enhance the migration ability of cervical cancer cells. SUV39H1 is down-regulated in a variety of leukemia, and knockdown of SUV39H1 leads to an increase in the number of leukemia stem cells (LSCs) and accelerates disease progression (Chu et al., 2020; Hansen et al., 2022). Interestingly, in ovarian cancer, high expression of SUV39H1 promotes the proliferation of cancer cells and leads to poor prognosis in patients (Li, Shao & Zhao, 2021). SUV39H1 is also believed to be involved in the development of liver cancer in a considerable number of patients infected with hepatitis B virus (Takeuchi et al., 2020). In human colon cancer (CRC), inhibition of SUV39H1 significantly increases the expression of granulozyme B, perforin, FasL, and IFN, inducing tumor cell apoptosis by enhancing immune surveillance and the killing effect of CTL (Lu et al., 2019). In prostate and nasopharyngeal carcinoma, decreasing SUV39H1 expression inhibit the proliferation and invasion of tumor cells (Lai et al., 2018; Yan et al., 2020). Our study shows that SUV39H1 has different transcriptional levels in different tumors. In OSCC, the expression of SUV39H1 is significantly increased, which is significantly related to tumor size, suggesting that SUV39H1 may play a role in promoting tumor development in OSCC.

Tumor size is closely related to tumor angiogenesis (Fukumura et al., 2018). In the absence of vascular support, tumors rarely develop beyond 2 mm3, demonstrating the critical role of angiogenesis in tumor growth and development (Fukumura et al., 2018). Notch1 controls angiogenesis by modulating endothelial cell fate determination during angiogenesis (Liu et al., 2003; Ye et al., 2023). A variety of angiogenic signaling pathways are regulated by Notch1, such as VEGF (Chen et al., 2022; Kangsamaksin, Tattersall & Kitajewski, 2014). NOTCH1 plays an important role in oral squamous cell carcinoma angiogenesis. So far, to the best of our knowledge, no studies on the link between SUV39H1 and NOTCH1 have been reported. Therefore, in order to clarify whether the promotion of tumor growth by SUV39H1 is related to tumor angiogenesis, we studied the correlation between the expression of SUV39H1, Notch1 and CD31 proteins, and the results showed that the expression of SUV39H1 was significantly positively correlated with the expression of Notch1 and CD31 proteins. Both our experimental results and those of GEPIA database show that the expression of NOTCH1 was positively correlated with that of CD31. This suggests that SUV39H1 may be involved in the angiogenesis regulated by Notch1.

Chaetocin is a nonspecific SUV39H1 inhibitor (Yang et al., 2022). However, a separate report showed that Chaetocin has also been identified as histone methyltransferase SUV39H1 specificity inhibitor (Guerra et al., 2021). In a recent study, Chaetocin was widely used as SUV39H1 inhibitors (Sak et al., 2021). In this study, we used the Chaetocin as SUV39H1 inhibitors. Our results show that the Chaetocin significantly inhibit the growth of OSCC, and no obvious side effects. Immunohistochemistry showed that Chaetocin significantly reduce SUV39H1, Notch1, CD31 expression. These results further suggest that inhibition of SUV39H1 may reduce tumor angiogenesis. Interestingly, in vivo experimental results showed that the expression of Ki67 was significantly reduced in the administration group, indicating that Chaetocin can significantly inhibit tumor cells proliferation. This is similar to our results in human specimens, where SUV39H1 expression correlates with tumor stage.

Considering that the target of drug intervention is non-specific, the RNAi technology used in the in vitro study was used to intervene the expression of SUV39H1. VEGF increase microvascular density and vascular permeability in tumor tissue. Tumor cell-derived VEGF is one of the important factors mediating tumor angiogenesis (Apte, Chen & Ferrara, 2019). Oral squamous cells were treated with SUV39H1 siRNA and/or DAPT, and conditioned culture-medium was collected. The migration and tube formion ability of HUVEC were significantly different in different medium conditions. SUV39H1 siRNA significantly decreased the expression of Notch1 and VEGF in OSCC cells. DAPT further decreased VEGF expression without affecting SUV39H1 expression. By replacing SUV39H1 siRNA with Chaetocin, we obtained similar results, partially demonstrating that Chaetocin can inhibit the action of SUV39H1. This part of the experimental results can partly prove the reliability of the results of animal experiments. These results suggest that SUV39H1 may regulate tumor angiogenesis through Notch1 in OSCC cells.

It should be noted that this study did not further investigate the specific mechanism by which SUV39H1 regulates Notch1 expression. We speculate that there are two possibilities. One is that SUV39H1 regulates Notch1 expression during gene transcription. In addition, as a transcription factor, the intracellular domain of Notch1 expression may have protein-protein interaction with SUV39H1. Chaetocin is a non-specific inhibitor of SUV39H1, which can only partially explain the effects of inhibiting SUV39H1 on tumor growth and intracellular molecular expression. For in vivo experiments, the use of Chaetocin cannot fully represent the effect of SUV39H1 knockdown on tumor biological behavior, which needs to be further verified by gene editing technology.

Conclusions

In summary, our findings suggest that inhibition of SUV39H1 in head and neck squamous cell carcinoma may affect angiogenesis by regulating Notch1 expression. This study is likely to strengthen interest in SUV39H1 as a potential prognostic and therapeutic target for OSCC.

Supplemental Information

Supplemental Information 1 Correlation between NOTCH1 and CD31 in the GEPIA database.

The expression of NOTCH1 was positively correlated with that of CD31.

Supplemental Information 2 The original data of the immunohistochemical experiment.

Immunohistochemical experimental data for statistical analysis.

Supplemental Information 3 Raw data from in vitro experiments.

The data were analyzed statistically in vitro.

Supplemental Information 4 The original picture of the in vitro experiment.

Supplemental Information 5 The original image of the in vivo experiment.

Supplemental Information 6 The original image of the immunohistochemical experiment.

Supplemental Information 7 ARRIVAL 2.0 check.

Additional Information and Declarations

Competing Interests

Author Contributions

Human Ethics

Animal Ethics

Data Availability

The authors declare that they have no competing interests.

Yan Chen conceived and designed the experiments, performed the experiments, analyzed the data, prepared figures and/or tables, authored or reviewed drafts of the article, and approved the final draft.

Xiuhong Weng conceived and designed the experiments, performed the experiments, prepared figures and/or tables, authored or reviewed drafts of the article, and approved the final draft.

Chuanjie Zhang conceived and designed the experiments, analyzed the data, prepared figures and/or tables, and approved the final draft.

Simin Wang conceived and designed the experiments, analyzed the data, prepared figures and/or tables, and approved the final draft.

Xuechen Wu conceived and designed the experiments, prepared figures and/or tables, and approved the final draft.

Bo Cheng conceived and designed the experiments, prepared figures and/or tables, authored or reviewed drafts of the article, and approved the final draft.

The following information was supplied relating to ethical approvals (i.e., approving body and any reference numbers):

This study was approved by the Medical Ethics Committee of Zhongnan Hospital of Wuhan University (Ethical Application Ref: WDRY2021-KS019) and was conducted in accordance with the guidelines of the Declaration of Helsinki on human experimentation.

The following information was supplied relating to ethical approvals (i.e., approving body and any reference numbers):

This study was approved by the Medical Ethics Committee of Zhongnan Hospital of Wuhan University (protocol code ZN2022085) and was conducted in accordance with the guidelines of the Declaration of Helsinki on human experimentation.

The following information was supplied regarding data availability:

The raw measurements are available in the Supplemental Files.

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
