# Peer review of "Inhibition of SUV39H1 reduces tumor angiogenesis via Notch1 in oral squamous cell carcinoma"

_PeerJ, doi:10.7717/peerj.17222_

## Round 0.1 · original submission · Major Revisions

Dear Dr. Chen,

Please make the required edits to your manuscript and resubmit.

With kind regards,
Jian Song

Reviewer 1 ·

Basic reporting

The overall language used in the manuscript is professional and satisfactory; however, there are some points of concern regarding other aspects of basic reporting. See specific comments below.
- While introducing SUV39H1, the authors should also detail its canonical functions in H3K9 methylation. The authors should also consider adjusting the term "first" in "the first human histone lysine methyltransferase" – this statement is misleading as it is neither first in terms of evolution nor the first methyltransferase discovered. Please revise or add additional context.
- It is recommended to check the appropriateness of all references cited in the introduction. Some potential issues: i.e., Line: 54-55: "regulating protein activity, stability, and protein-protein interactions [4, 5]" – please confirm the validity of these claims and the references cited. Missing references for lines 57-58: "SUV39H1 may be a tumor suppressor, involved in the occurrence and development of various types of tumors by promoting cell senescence and inhibiting genes required for cell proliferation". Lines 63-64: "Angiogenesis is closely related to tumor size, which does not exceed 2mm3 in the absence of new blood vessels[11]." – from a brief look, the cited article does not appear to provide direct support to this claim nor provide the 2mm3 size.
- Consider referring to Fig 1C in line 169.
- Figure 2 captions: please specify how the protein level quantifications were done. The main text mentioned "Tissue Score", but the figure shows "Expression" or "Histoscore". Please adjust and indicate the quantifications used in the captions. Also, in Figure 2: the terms "Expression of NOTCH1" or Expression of SUV39H1" could easily be misunderstood as gene expression at the transcriptome level. I suggest using another term that may be more appropriate in the context of IHC – like staining intensity or tissue score – depending on how the qualifications were done. Please check and adjust as necessary.
- Figure 3A: please indicate what the image on the top and bottom represent in the captions. Also, indicate at what time points these images were taken and add annotations (arrows, etc.) to specific characteristics of interest (if applicable).
- Figure 3C: using both upper and lower error bars is recommended.
- Please provide the raw numeric data of Figures 3D and 3E in the supplemental files. Also, indicate the statistical test used in the Figure caption to provide the necessary context. Whenever possible, please use both upper and lower error bars as they give more transparency on data variability.
- Figure 5C: it is recommended to provide representative images of the migration assay.
- The link between NOTCH1 and SUV39H1 was not sufficiently introduced. It would help to describe how the reported correlation was initially discovered.
- Please check and properly label all raw data in the supplemental files. Particularly the images within the files containing raw images.
- References and in-text citations need to be adjusted to PeerJ style.

Experimental design

There are some significant limitations to the experimental design. Below are some specific comments.
- Line 183 "Chaetocin is used to inhibit SUV39H1 function". There are other targets inhibited by Chaetocin aside from SUV39H1 – i.e., Thioredoxin Reductase, HIF-1α, HSP90 (see PMID: 34464601). The conclusions from the Chaetocin are only valid to the extent that the compound itself could inhibit the growth of OSCC in vivo and cannot be used to link SUV39H1 due to its off-targets. To confirm the role of SUV39H1, the authors should consider genetic approaches, such as knockdown or overexpression studies.
- Figure 3 was primarily composed of in vivo studies performed with Chaetocin treatment and CAL27 xenografts. The results could be validated by a knockdown of SUV39H1 in the CAL27 cell line studied to monitor for antiproliferative effects in vitro. It would also be of interest to check for the decrease of NOTCH1 in the CAL27 cells (used in the in vivo xenograft).
- In Figure 3-4, the in vivo studies were performed with Chaetocin treatment, while the in vitro studies, in Figure 5, were performed with siRNA knockdown of SUV39H1. It may be of interest to also perform Chaetocin treatment in the experiments in Figure 5 to investigate if similar phenotypes could be observed.
- Two siRNAs were validated in Figure 5A, but only one was used in Figure 5B-D. Additional sources of validation, either by an additional SUV39H1 siRNA target sequence or overexpression experiment, are recommended.
- It is recommended to show validation of the robustness of the IHC-based protein level determination. Preferably via an independent method like a western blot on a subset of samples with corresponding IHC data, if resources/materials permit.

Validity of the findings

There are some significant limitations to the validity of the findings. Below are some specific comments.
- Figure 2B, 2C, although they show statistically significant results, the correlation coefficient presented (r, being less than 0.3) is typically interpreted as a weak correlation. Although the results are statistically significant, the practical significance could be limited due to the weak correlation. I would recommend being more careful with the interpretation of these results to avoid potential overinterpretation of their significance.
- Section starting line 190: "Chaetocin inhibit tumor angiogenesis in OSCC" – this is not well supported. The result does not clearly demonstrate a decrease in tumor angiogenesis.
- Chaetocin is not a sufficiently selective inhibitor to SUV39H1 to demonstrate that SUV39H1 inhibition led to the decrease in tumor growth in vivo. See the related comment above in the Experimental Design section.
- Chaetocin inhibits SUV39H1, and it isn't generally associated with reductions in the protein levels of SUV39H1. Therefore, the observed decrease in SUV39H1 level post-Chaetocin treatment (Figure 4B) is an interesting piece of data that warrants further investigation. To confirm this finding and strengthen the reliability of the data, it would help to perform a western blot to look at SUV39H1 levels. This could be done on a selected set of xenograft tumor samples previously treated with Chaetocin (if available) or by treating CAL27 cells for a similar duration with Chaetocin. This additional data would help substantiate the claim that Chaetocin reduces SUV39H1 protein levels.
- The authors suggest that the reduction in tumor angiogenesis is via a reduction in NOTCH1 following inhibition/knockdown of SUV39H1. To substantiate the claim, it is recommended that a NOTCH1 knockdown be performed and that the Transwell and tube formation assays (as done in Figure 5B) be repeated with knockdown NOTCH1. Alternatively, overexpression of NOTCH1 with SUV39H1 knockdown could be performed to observe whether the same phenotypes from the SUV39H1 knockdown are reduced. These experiments could provide more robust evidence for NOTCH1's involvement.
- Only one siRNA target sequence was used in the experiments presented in Figure 5B-E. It is recommended that 2-3 target sequences be used for validation of the findings from siRNA knockdowns.

Additional comments

The study presents findings of interest, but there are notable limitations in the experimental design and validity of findings that should be addressed. Specific comments are noted above. It is also recommended to have more detailed discussions and provide more context for each piece of data presented, as it can strengthen the narrative of the work.

·

Basic reporting

The article was expertly crafted, guaranteeing effortless understanding and a coherent, rational progression that facilitated seamless comprehension.

Experimental design

This study exhibits a meticulous and well-structured design.

Validity of the findings

The results presented in this study effectively tackle the fundamental research question at hand.

Additional comments

Minor concerns:
1. Since the paper does not include the overexpression of SUV39H1 experiment, it would be appropriate to remove the conclusion statement: "Our results suggest that overexpression of SUV39H1 promotes tumor development in OSCC."
2. The analysis of NOTCH1 and CD31 in the GEPIA database is recommended to determine their significance in normal and tumor tissues of head and neck squamous cell carcinoma.
3. Whether the knockdown of SUV39H1 in vitro leads to a reduction in cell proliferation of tumor cells and vascular endothelial cells?
4.Whether the in vitro treatment of chaetocin results in a decrease in cell proliferation of tumor cells and vascular endothelial cells. Also should include Ki67 staining for in vivo.
5. In Figure 5A, there is a mislabeling of the western blot. The positions of "Scramble" and "SUV39H1 siRNA" should be exchanged.
6. To elucidate how the knockdown of SUV39H1 decreases the levels of NOTCH1 and CD31, it is important to consider experiments involving transcription activity and the effects of H3K9me3 inhibition on the removal or direct modification of NOTCH1 and CD31. These experiments should be conducted and thoroughly discussed to gain a comprehensive understanding of the underlying mechanisms.

Reviewer 3 ·

Basic reporting

Clear scientific reporting

Experimental design

Method is clear, hard to comment on rigor

Validity of the findings

reasonable finds

---

## Round 0.2 · Major Revisions

See the comments from Reviewer 1. The authors need to highlight the importance of any missed changes, and explain that they must be addressed before publication. Importantly, the authors need to clarify which reviewer comments should/should not be addressed.

Reviewer 1 ·

Basic reporting

The authors addressed some of the comments in basic reporting from the last round of review, but there are still concerns that need further attention.
- Line 179: “Tissue scores of SUV39H1, Notch1 and MVD were calculated” – it is unclear how this score was calculated or if it was robustly determined (i.e., with blinding, etc.). It would help to contextualize how this quantification was done in this study.
- Figure 5C was not updated. No representative wound healing assay image was shown. Wound healing assay raw images are also not provided in the supplemental. Please clarify if both wound healing assay and Transwell migration were performed, as they are distinctly different experiments.
- Please disclose the concentration of Chaetocin and DAPT used for each cellular experiment and how long the course of the treatment was. This is important for evaluating the on-target effect and ensuring the experiment's reproducibility. It is also necessary to describe how the siRNA transfection was performed and how long post-transfection was the phenotypes observed.
- For all plots, it is recommended that the figure captions indicate the number of replicates performed, whether they are technical or biological replicates, what statistical test was performed (if applicable), and what the error bars represent (if displayed).
- Supplemental Table 1 appears to contain the same information as Table 1.
- Significant language edits are still needed to improve clarity. Specifically, the Methods (and supplemental methods) sections are difficult to follow.

Experimental design

- Generally, one siRNA target sequence is insufficient to draw robust conclusions. Considering the importance of the knockdown data in this manuscript, it is highly recommended to provide, at minimum, another validation (i.e., another siRNA target sequence) or an orthogonal experiment (i.e., overexpression experiment showing the opposite effect, etc.).
- The Methods section was confusing in some instances. i.e., the section on “Wound-healing assay and tube formation assay”. The main text describes CAL27 cells being used and referred to the supplemental methods, but the supplemental methods describe HUVEC cells. It wasn’t clear which cell type was used for what experiment. Also, the supplemental methods on the wound healing experiment described “cells migrated into the gap were counted”, but it wasn’t clear how exactly this counting was done, nor were there any images provided. It is highly recommended to ensure that all methods used in this study are described adequately, with sufficient information for the scientific community to independently reproduce the experiments.
- It is recommended to integrate the supplemental methods into the main Methods section to avoid unnecessarily referring to supplemental documents.
- The source/vendor of the siRNA or the target sequences should be included. Please also check that information or source of all other reagents used are provided.

Validity of the findings

Despite the first round of revision, there are still numerous significant deficiencies in the validity of the findings. Specific comments are as follows:
- Although the correlation has a statistically significant p-value, the magnitude of the correlation (R score) was quite weak (less than 0.3) in Figure 2B-C. In the last round of review, a suggestion was provided to avoid overinterpreting the data in Figure 2B-D and to discuss the data more holistically. The authors agreed, in the response letter, that “In view of the weak correlation between the two proteins, we have adjusted the interpretation of the results in this paper”. The current version of the section of the manuscript describing these panels seems to have very minimal changes. Please check lines 175-185 and revise as appropriate.
- In the response letter, the authors mentioned: “We investigated the expression of SUV39H1, Notch1 and VEGF in tumor cells treated with Chaetocin and DAPT and found that Chaetocin could decrease the expression of SUV39H1, Notch1 and VEGF as shown in figure5” – while the authors showed similar effects through western blotting, the phenotypic effect (i.e., experiments on Fig. 5B-D) was not demonstrated with Chaetocin treatment. It is recommended to demonstrate the Chaetocin-induced phenotypes in vitro to compare them to the SUV39H1 knockdown effects to make the argument more convincing. Considering that the relevance of SUV39H1 in the in vivo studies was not fully justified (due to the non-specificity of the inhibition of SUV39H1 by Chaetocin), the limitations of the study should be thoroughly discussed.
- In the response letter, the authors noted, “As shown in Figure 1D, the expression of SUV39H1 is related to the T stage of the tumor, so we speculate that the decrease of SUV39H1 expression can inhibit the proliferation of tumor cells” – it was not clear in Figure 1D whether there is a significant correlation. There also do not appear to be any statistical tests performed. Furthermore, this appears more speculative than based on empirical data.

Additional comments

Overall, the current manuscript presents data suggesting that Chaetocin inhibits tumor growth and angiogenesis in vivo and data suggesting that SUV39H1 knockdown affects NOTCH1/VEGF and angiogenesis in vitro. With potentially weak evidence linking SUV39H1 to the observed phenotype from Chaetocin in vivo, considering the non-specific nature of Chaetocin-mediated inhibition of SUV39H1, the conclusions should not be overstated, and a thorough discussion of the limitations of this study is highly recommended.

·

Basic reporting

no comment

Experimental design

no comment

Validity of the findings

no comment

Additional comments

The substantially revised manuscript has adequately addressed all reviewer comments.

---

## Round 0.3 · Minor Revisions

Hereby I confirm that the authors have addressed most of the reviewers' comments and this manuscript is almost ready for publication.

Please address the remaining items from Reviewer 1

Reviewer 1 ·

Basic reporting

The authors have addressed most of the comments from the previous round of review satisfactorily.
- Please make sure the Chaetocin concentration used is reported. The rebuttal letter indicates that it was reported in the Figure caption, but could not be found.
- An appropriate statistical test on Figure 1C-D may help clarify the reported observed effects.

Experimental design

No additional comments.

Validity of the findings

No additional comments.

---

## Round 0.4 · accepted · Accept

I confirm that the authors have addressed all of the reviewers' comments.